# Evaluating marine dust records as templates for optical dating of Oldest Ice

Jessica Ng[1], Jeffrey Severinghaus[1], Ryan Bay[2], and Delia Tosi[3]

[1]Scripps Institution of Oceanography, University of California San Diego, La Jolla, CA, 92093, USA
5 [2]Department of Physics, University of California Berkeley, Berkeley, CA, 94720, USA
[3]Department of Physics, University of Wisconsin Madison, Madison, WI, 53706, USA

*Correspondence to*: Jessica Ng (jyn002@ucsd.edu)

**Abstract.** The continuous ice core record extends 800,000 years into the past, covering the period of 100,000-year glacial cycles, but not the transition from 40,000-year glacial cycles (the Mid-Pleistocene Transition, 1.2-0.7 million years ago). A 10 primary goal of the International Partnerships in Ice Core Sciences is therefore to retrieve a 1.5-million-year-old continuous ice core, increasing our understanding of this major change in the climate system and thus of fundamental climate forcings and feedbacks. However, complex glacial processes, limited bedrock data, and young basal ice in previous cores necessitate careful reconnaissance studies before extracting a full core.

15 Ice borehole optical logging reflects the ice dust content and may be used to date ice quickly and inexpensively if a reference record is known. Here we explore the relationship between ice dust records and well-dated marine dust records from sediment cores in the southern Atlantic and Pacific Oceans, which lie along paths of dust sources to Antarctica. We evaluate how representative these records are of Antarctic dust both through the existing ice core record and during the older target age range, suggesting that a newly published 1.5 million year record from site U1537 near South America is likely the most robust 20 predictor of the Oldest Ice dust signal. We then assess procedures for rapid dating of potential Oldest Ice sites, noting that the ability to detect dating errors is an essential feature. We emphasize that ongoing efforts to identify, recover, date, and interpret an Oldest Ice core should use care to avoid unfounded assumptions about the 40 kyr world based on the 100 kyr world.

## 1 Introduction

The marine sediment record shows that the climate system cycled through glacial and interglacial states with regular ~41,000 25 year (41 kyr) periodicity prior to 1.2 million years ago (Ma) in the so-called 40k world (Figure 1) (Lisiecki and Raymo, 2005). Between 1.2 Ma and 0.7 Ma, the periodicity of these cycles lengthened and became less regular while their magnitude increased, an event known as the Mid-Pleistocene Transition (MPT). Finally, from 0.7 Ma until the present, irregular ~100 kyr cycles have characterized the climate system, as seen in both marine sediment and ice core records.

Our understanding of the MPT and the 40k world remains limited in large part because the continuous ice core record extends only 800 kyr into the past (Lambert et al., 2008), with a few discrete snapshots of older climate up to 2.7 million years old (Yan et al., 2019). A central question is *why* the transition happened, with implications for the fundamental mechanisms of the climate system; see Berends et al. (2021) for a thorough review. Clark et al. (2006) proposed that soft regolith eroded over repeated glaciations in the early Pleistocene, eventually exposing unweathered bedrock that allowed thicker ice sheets to grow and introduce nonlinear feedbacks. Tectonic activity may have gradually lowered atmospheric $CO_2$ and led to more extreme glacial cooling (Raymo et al., 1988; Ruddiman and Raymo, 1988; Legrain et al., 2023); on the other hand, Honisch et al. (2009) question the role of $CO_2$ based on evidence that its atmospheric concentration during 40k-world interglacial periods did not change, a finding supported by snapshots of the early Pleistocene atmosphere from blue ice cores (Yan et al., 2019). McClymont et al. (2013) interpret gradual global decrease in SSTs across the MPT as evidence of a change in internal climate feedbacks. Chalk et al. (2017) suggest that both changing ice sheet dynamics and carbon cycle feedbacks played a role, while Willeit et al. (2019) emphasize the importance of both declining $CO_2$ and regolith removal. In a number of these cases, an abrupt increase in the duration of glacial cycles may have resulted from frequency locking of a slowly increasing internal period to insolation variations (Nyman and Ditlevsen, 2019). Beyond the driver of the MPT, related questions regarding the early and mid-Pleistocene include the history of Antarctic temperature before 800,000 years ago (800 ka) and the role of carbon dioxide in large ice sheet dynamics (Fischer et al., 2013).

A 1.5 Ma continuous ice core would greatly advance our understanding of Quaternary climate and has been stated as a goal of the International Partnership in Ice Core Sciences. Several groups worldwide are already taking steps toward drilling an Oldest Ice core (Fischer et al., 2013; Goodge and Severinghaus, 2016; Van Liefferinge et al., 2018; Zhao et al., 2018). These efforts must contend with the unprecedented challenges of retrieving 1.5 Ma ice: the geothermal heat flux of the bedrock must not be high enough to melt the basal ice containing the oldest ice; the ice sheet must not be so thick that it insulates the geothermal heat and melts the basal ice; and the bedrock topography must not have disturbed the stratigraphy (Fischer et al., 2013; Chung et al., 2023).

Given the sparse coverage of geothermal heat flux and bedrock topography data across Antarctica, efforts to drill an Oldest Ice core require extensive reconnaissance work to increase the likelihood of selecting drill sites with intact 1.5-million-year-old basal ice. Several efforts are pursuing "rapid access drilling," a reconnaissance mode of exploration in which boreholes are quickly drilled or melted through the ice sheet without retrieving intact ice cores (Goodge and Severinghaus, 2016; Winebrenner et al., 2013). Rather than 5 or more years, these techniques require less than a week to reach the base of the ice sheet and recover short basal ice samples (Goodge et al., 2021). Though most of the ice is not retrieved, indirect measurements such as borehole temperature and dust content may be made in conjunction with rapid access drilling. The dust content of ice is measured by optical logging, in which a light source is lowered through the borehole and the dust content of the ice

determines the amount of light backscattered (Bay et al., 2001; Chan et al., 2017; Goodge et al., 2021). This technique allows for rapid dating of the ice by comparison to a well-dated reference record.

While no Antarctic dust record exists beyond 0.8 Ma to serve as such a reference, the dust record from marine sediment cores around Antarctica extends as far back as 4 Ma (Martínez-Garcia et al., 2011) and has been proposed as a stratigraphic template for dating Oldest Ice (Wolff et al., 2022). With the exception of minor, intermittent contributions from local volcanos, the dust measured in Antarctic ice originates from Southern Hemisphere terrestrial sources, primarily South America, Australia, and

New Zealand (Neff and Bertler, 2015). Some is deposited in the oceans and archived in marine sediments in the Southern Ocean before it reaches Antarctica. During glacial periods, exposure of continental shelves and reduced washout due to weakened rainfall as well as increased gustiness and aridity drives higher dust transport and deposition to both subantarctic marine sediments and Antarctic ice (McGee et al., 2010; Markle et al., 2018). Ice cores from across Antarctica appear to have a common dust (or calcium) signal through time, suggesting the use of dust as an ice dating tool (Baggenstos et al., 2018;

Mulvaney et al., 2000).

Previous work relied on a single marine sediment core from site ODP 1090 in the southern Atlantic Ocean and focused on its similarities to the ice core dust signal during the past 800 kyr (Martínez-Garcia et al., 2011; Wolff et al., 2022). However, the critical time period for using marine dust to date an Oldest Ice core is 800 ka – 1.5 Ma, i.e., before and during the MPT when

the climate system is known to have been different. Here we build on the concept of marine dust as a stratigraphic template for Oldest Ice by comparing the ODP 1090 dust record to a new higher-latitude 1.5 Ma marine dust record from the southern Atlantic Ocean and benthic $\delta^{18}$O (Wolff et al., 2022; Weber et al., 2022; Lisiecki and Raymo, 2005). We examine changes in the agreement of these two records from the 40 kyr world to the 100 kyr world. Finally, we demonstrate how marine dust records can be used with rapid drilling and optical dust logging for Oldest Ice site reconnaissance, and we suggest other visual

pattern-matching strategies that may assist in dating these sites in conjunction with rapid access drilling and borehole optical logging.

## 2 Data description

As templates for the Oldest Ice dust record, we use the two oldest subantarctic marine dust records and the global benthic $\delta^{18}$O stack and compare them to the two oldest continuous Antarctic ice core dust records (Figures 1-2).

### 2.1 EPICA Dome C (EDC)

The European Project for Ice Coring in Antarctica Dome C (EDC) ice core was drilled from 1993 to 2004 at 75.1 °S, 123.35 °E and covers the past 800 kyr (Lambert et al., 2008). Dust content was measured by both Coulter counter and laser sensor, which agreed well; we use the laser data, as they are measured at higher resolution. We use the AICC2012 age scale, which

improves upon the original EDC3 chronology with additional ice and gas stratigraphic markers, new atmospheric $\delta^{18}O$ measurements for orbital constraints, and an improved inverse dating tool (Datice) that balances various chronological information to produce a common timescale for multiple Antarctic ice cores (Veres et al., 2013; Bazin et al., 2013)

### 2.2 Dome Fuji (DF)

The Dome Fuji (DF) ice core was drilled at 77.31 °S, 39.70 °E in two sections: the first in the 1990s representing the past 340 kyr and the second in the 2000s ending at 720 ka. We use the first record until the second record begins at 296 ka, when we switch to the second record because of its higher time resolution. Dust microparticles were measured with a laser particle counter. The DFO2006+AICC2012 age model is based on aligning isotopic records to EDC records on the AICC2012 chronology (Dome Fuji Ice Core Project Members: et al., 2017; Fujii et al., 2003).

### 2.3 Scotia Sea, International Ocean Drilling Program Site U1537 (U1537)

A new 1.5 million year old record of magnetic susceptibility as a dust proxy was recently published from a marine sediment core drilled in 2019 at 59.11 °S, 40.91 °W in the Scotia Sea in the southern Atlantic Ocean (Weber et al., 2022). Close resemblance to Antarctic non-sea salt $Ca^{2+}$ at high resolution has shown that magnetic susceptibility of marine sediments in the Scotia Sea is a reliable indicator of dust transported by the atmosphere without major influence of marine transport from sea ice, icebergs, or ocean currents (Weber et al., 2012). The age scale was established by synchronizing magnetic susceptibility to the EDC dust flux on the AICC2012 time scale from 0-800 ka and to the LR04 benthic $\delta^{18}O$ stack from 800 ka – 1.5 Ma, with additional tie points based on magnetic reversals.

### 2.4 Southern Atlantic Ocean, Ocean Drilling Program Site 1090 (ODP1090)

A 4 million year old marine sediment core was drilled in 1997 at ODP site 1090 in the southern Atlantic Ocean at 42.91 °S, 8.90 °E (Martínez-Garcia et al., 2011). The dust accumulation rate shown here is very similar to the iron accumulation rate used in Wolff et al. (2022). We have transferred the age scale for the past 800 kyr, which is based on visual correlation of X-ray fluorescence iron measurements as a dust proxy to the EDC dust record, to the AICC2012 chronology (Veres et al., 2013; Bazin et al., 2013). The original ODP Site 1090 age model is used for the period 800 ka to 2.9 Ma based on alignment of benthic $\delta^{18}O$ to the LR04 benthic $\delta^{18}O$ record, which in turn is tuned to orbital forcing.

### 2.5 LR04 Benthic Stack (LR04)

While benthic $\delta^{18}O$ is not a direct proxy for Antarctic dust, the timing of glacial-interglacial fluctuations is common between the two as the growth and decay of ice sheets regulates both 1) the $\delta^{18}O$ and temperature of the global oceans, which determines the $\delta^{18}O$ of calcite of benthic foraminifera, and 2) the aridity (due to temperature) and circulation of the atmosphere, which influences the production, transport, and atmospheric lifetime of dust. The 5.3 million year old LR04 stack includes 57 globally

distributed benthic $\delta^{18}O$ records aligned using an automated graphic correlation program (Lisiecki and Raymo, 2005). The LR04 age model was tuned to a simple model of ice volume based on summer insolation at 65 °N.

## 3 Comparison of dust records

To assess the degree to which the dust signal in marine sediments represents an Oldest Ice dust signal, we first compared the dust records over the common time period, 3.3 – 715.9 ka (Table 1). We sampled the dust records at common 100 year timesteps, normalized them to unitless values between 1 and 10, and correlated each pair of records. The strong correlation between DF and EDC ice dust (R = 0.83) indicates that the dust signal is consistent across the East Antarctic Plateau. The correlation between the marine dust records is also strong over this period (R = 0.78), suggesting that the marine dust signal is consistent throughout the mid-to-high latitude southern Atlantic Ocean after the MPT. Each ice record matches with each marine dust record to a similar degree, with correlation strengths falling within the range of 0.72-0.85 (Table 1). We also compared the dust records to the LR04 benthic $\delta^{18}O$ signal using the logarithm of the dust records to represent the nonlinear relationship between glacial conditions and dust flux (Shaffer and Lambert, 2018). The relationship between LR04 and EDC, U1537, and ODP1090 is strong over the common time period (R = 0.73-0.76), while the correlation with DF is slightly weaker (R = 0.64).

However, the time period in which marine records are actually required to extend the existing ice dust record is 800 ka – 1.5Ma, across the MPT and into the 40k world, when the climate system is understood to be different from the 100k world. The new U1537 record allows us to examine the robustness of the marine dust record into the pre-MPT period. A rolling correlation with a 200 kyr window demonstrates that the relationship between U1537 and ODP1090 weakens beyond 800 ka, the target Oldest Ice period (Fig. 3, R = 0.34 over 800 ka – 1.5 Ma). On the other hand, U1537 dust continues to agree with LR04 benthic $\delta^{18}O$ over this time period (R = 0.66).

The disagreement between marine dust records raises the question of which to use for dating Oldest Ice. We suggest that U1537 is likely more representative than ODP1090 of Oldest Ice dust. U1537 is located much closer to both Antarctica and southern South America, one of the primary sources of glacial dust to East Antarctica (Delmonte et al., 2008; Li et al., 2008). In contrast, ODP 1090 is located about 11,000 km further east and nearly 20° further north (42.91 °S), in the mid-latitudes rather than high latitudes (Fig. 2). U1537 is also much more highly resolved, with 200 m covering 1.5 Ma in the high deposition Scotia Sea compared to 40 m at ODP1090. Based on these criteria and discussed further in Sect. 5, we suspect that ODP1090 and U1537 are responding to latitudinally variable climate signals, and that U1537 more closely represents the Antarctic dust pattern. On the other hand, the prevalence of Patagonian dust observed at EDC over the past 800 ka (Delmonte et al., 2008) could change further back in time, in which case ODP1090 could better represent integrated dust emissions from across South America. Still, the quantity—if not the geochemistry—of dust transported to both U1537 and east Antarctica would likely

remain similar unless dust emissions from Patagonia were quite different from the rest of the continent. While a composite record could be developed incorporating both ODP1090 and U1537, it would likely decrease the predictive ability of U1537 for dating Oldest Ice. The close agreement between LR04 and U1537 suggests a relationship between ice volume, temperature, and dust continuing into the 40k world; however, as benthic $\delta^{18}O$ is not a direct comparison to dust, we refrain from using LR04 as a dust template to avoid the circularity of assuming this relationship when using the correspondence between records to interpret climate signals. We will use U1537 to represent the marine dust record throughout the rest of this study.

### 3.1 Improving visual pattern matching

Our approach relies on visual pattern-matching that takes advantage of the common pacing between the sequence of peaks in the marine and ice dust records. While the glacial-interglacial pattern of dust is consistent between EDC and U1537 on established age scales, some dust peak tie points may be ambiguous, and differences in shorter-term variability may make identifying common peak shapes more difficult.

We briefly explore a collection of techniques that may aid in visual pattern-matching (Fig. 4, Table 2). These include using the logarithm of the records (Fig. 4d-f), smoothing with a 20 kyr running mean (Fig. 4b and e), and taking the derivative of the smoothed records (Fig. 4c and f). The logarithmic EDC and U1537 dust records were more strongly related than the original normalized records, as the logarithm reduces the difference in dynamic range, e.g., at the penultimate glacial maximum around 150 ka (R = 0.87 compared to 0.83). Smoothing increased the correlation strength between U1537 and EDC for both normalized (R = 0.91) and logarithmic records (R = 0.93); however, we caution that smoothing over time relies on an initial age scale constructed for a dust-depth record, and over-smoothing may dampen the distinctive peak shapes that are used for pattern-matching. The derivatives of the smoothed and logarithmic smoothed records have higher frequency variability that make them sensitive to offsets between records. This feature could be useful for fine millennial scale alignment, but it may distract from the glacial-interglacial variability that is the focus of dating for Oldest Ice reconnaissance. Using a combination of multiple techniques to visually identify and match tie points could help avoid mismatched tie points from any single approach and produce a more robust estimate of the basal ice age.

### 4 Application to Rapid Dating of Oldest Ice Sites

### 4.1 Optical dust logging

Optical logging methods have been developed over the past few decades to collect data from glacial boreholes without the need to retrieve or destroy ice core samples (Bay et al., 2001). For example, at the South Pole, optical logging of IceCube Neutrino Observatory boreholes was used to produce a detailed dust record before the South Pole Ice Core was completed, showing close resemblance to high resolution non-sea-salt calcium measurements of the EDC ice core over the past 100 ka (The IceCube Collaboration, 2013). The logger is a ~1 m long instrument that shines a laser into the ice as it is lowered through

the borehole and measures the backscattered light. Brush baffles sweep away debris and prevent light from traveling directly through the borehole to the photon detector, ensuring that all measured light has travelled through the ice and interacted with impurities of interest. The logger measures a greater volume of ice than is possible with core measurements, often allowing for greater stratigraphic coherence of the dust record, although optical data is also susceptible to degradation from contamination of the drilling fluid and blemishes on the borehole surface.

The EDC borehole was optically logged in January 2010 and is shown to the base of the borehole for the first time here (Fig. 5). The entire borehole was logged in a single day with logging speed of ~25 cm s$^{-1}$ over a ~6 hour round trip, demonstrating the utility of optical logging for rapid dating and reconnaissance (The IceCube Collaboration, 2013). Depths were reconstructed from a combination of pressure sensor data, winch payout, and cable elongation corrections to an accuracy of 1-2 m and were used to transfer the record to the AICC2012 age scale. The optical log demonstrates very similar features to the conventional laser dust measurements of the EDC ice core (Fig. 5), which have previously been analyzed in detail (Lambert et al., 2008, 2012). Bubbly ice above 1250 m interacts with the logger light source differently from clear ice and is not shown here, but agrees closely with optical dust logs at the South Pole (The IceCube Collaboration, 2013).

## 4.2 Simulating Oldest Ice reconnaissance

Here we demonstrate how a potential Oldest Ice site could be dated using a suite of artificial optical dust records and visually aligning them to the U1537 marine dust record (Figs. 6-7). This simplistic approach is intended for rapid dating using only the dust record to estimate the basal age of the ice sheet, possibly in the field where decisions must be made quickly; more complex methods and other age data should be used to establish a detailed chronology once an Oldest Ice core is recovered.

We created four artificial records by modifying U1537 to mimic the common timing but different amplitudes between marine and ice dust peaks, including processes that affect the ice dust record (Fig. 6, Table 3). Artificial dust records 1 and 2 were created by smoothing the U1537 dust record with a 20 kyr running mean, re-scaling the smoothed record by random factors at 500 kyr intervals, and adding random noise at 2 kyr intervals. The resulting record was normalized between 1 and 10. For artificial record 1, the amplitude of the noise was adjusted such that the artificial record correlated with the original U1537 record with R = 0.71 (Table 3), slightly lower than the correlation strength of U1537 with DF and EDC over the past 800 kyr (Table 1). For artificial record 2, the amplitude of the signal from 800 ka – 1.5 Ma was reduced to mimic the observed decrease in amplitude of the EDC laser dust and optical dust log deeper in the ice sheet and further back in time (Fig. 5). While this decrease is also seen in U1537 between 500-800 ka (Fig. 1) and may not imply a different relationship between marine and ice dust, it is also possible that less extreme glacial periods in the 40k world could reduce dust transport to the ice sheet more than to marine sediments, or that depth-dependent processes in the ice sheet affect the strength of the optical dust signal. The correlation of the entire record with U1537 (R = 0.62) is weaker than either the 0 – 800 ka (R = 0.82) or 800 ka – 1.5 Ma (R =

0.75) period because of the reduced amplitude in the latter half, with the correlation of the older half somewhat weaker than the younger half (Table 3). Artificial record 3 was created by re-scaling U1537 by random factors at 100 kyr, 40 kyr, 10 kyr, 5 kyr, and 1 kyr intervals and reducing the amplitude beyond 800 ka, with a similar pattern of correlation strength for the younger half, older half, and entire record as artificial record 2. Artificial record 4 is identical to artificial record 3 except the period 1-1.3 Ma, which is reversed to mimic folded ice, resulting in a somewhat weaker correlation with U1537 over the older half of the record (R = 0.72 for artificial record 4 compared to R = 0.83 for artificial record 3, Table 3).

To simulate an optical log at an Oldest Ice site optical log where the basal age is unknown, we put the artificial records on a depth scale using the simple Nye 1D ice flow model (Fig. 7a) (Dansgaard and Johnsen, 1969; Nye, 1963). The process is shown for artificial record 1 in Fig. 7 for illustrative purposes; we used the same depth and age scale for the other artificial records as well. We used an ice sheet thickness of 3500 m and accumulation rate of 0.05 m yr$^{-1}$ to produce an end depth of 3152 m, approximating the EDC depth of 3260 m with substantial compression of the dust record at depth. Then we created an initial age scale for the artificial dust-depth record by using the Nye model in reverse with a different, more realistic ice thickness (3300 m) and accumulation rate (0.02 m yr$^{-1}$; Fig. 7b, top). The dust record was sufficiently stretched to identify distinct peaks, which we visually matched to the U1537 marine dust record with 24 tie points (Fig. 7b, gray lines). The techniques from Sect. 3.1 would assist in identifying and matching tie points at this stage. We linearly interpolated the age between these tie points to produce a revised age model, which in this exercise returns the artificial record to its original age model (Fig. 5c, "Manual").

We also explored the use of a dynamic time warping (DTW) algorithm, which has previously been used to align other optical logs and may be considered as a tool for dating Oldest Ice (The IceCube Collaboration, 2013). With the start and end tie points assigned, DTW calculates the Euclidian distance between corresponding points of two input time series and stretches the time series by repeating points to minimize the sum of distances (Micó, 2022). For artificial record 1, we used the start and end tie points in Fig. 7b (thick gray lines) as inputs to the DTW algorithm. Though the start and end tie points matched the original ages of the artificial record, the DTW algorithm misaligned peaks in the period from 550 ka to 800 ka such that peaks in the artificial record appeared younger than they originally were (Fig. 7c, "DTW"). These errors do not affect the end age of the DTW alignment, which is assigned as an input, but they do raise the question of how to evaluate the accuracy of optical log age models and ultimately the basal ice age.

The ability to identify dating errors is important for dating potential Oldest Ice sites, where ambiguities may arise in aligning ice and marine dust peaks, especially in deep, old, highly compressed ice. We investigated how incorrect alignment of tie points affects the correlation strength of U1537 with the artificial records dated using both manual and DTW approaches described above. To make the ages of the artificial record too young by one glacial cycle from approximately 1.25 Ma to the end, we mismatched the tie points starting at number 15, incorrectly matching artificial dust tie point 16 to U1537 tie point 15,

artificial dust tie point 17 to U1537 tie point 16, and so forth ( Fig. 7c, "Manual -1"). This offset weakened the relationship between the dated artificial record and U1537 from R = 0.72 to R = 0.64 (Table 4). We additionally neglected tie point 18 from the artificial record and shifted the subsequent tie points accordingly to produce an end age for the artificial record that is too young by two glacial cycles, which slightly weakened the correlation further to R = 0.61 (Table 4; not shown in Fig. 7c). To offset the artificial record such that it appears too old, we disregarded one (tie point 15) to two (tie points 15 20) tie points from U1537, resulting in a weaker correlation than the correct alignment (R = 0.65 and 0.67, respectively, compared to 0.72; Table 4, Fig. 7c). Overall, in this manual alignment approach, the correlation strength was highest for the correct alignment and decreased with misalignments. This pattern was consistent for all four artificial records (Table 4, rows 1-4) as well as for the alignments between the logarithmic artificial and U1537 records (Table 4, rows 5-8).

Finally, we applied DTW with offset end tie points that were 1-2 glacial cycles too old or too young. Because DTW adjusts the time series to minimize the disagreement between records, effectively optimizing the correlation, the relationship between the dated artificial record 1 and U1537 hardly varied with incorrect end tie points, with R constrained to 0.96-0.97 (Table 4, last row). Thus, dating errors were apparent for manually aligned peaks, but obscured when using DTW. Previous applications of DTW to align optical dust logs utilized other ice dust records with a high degree of similarity as stratigraphic templates (Bay et al., 2001, 2010). Rather than establishing a rough age model by alignment to marine dust, DTW may be useful for aligning optical logs of multiple boreholes across an Oldest Ice area.

## 5 Discussion

Ensuring that an Oldest Ice site will yield 1.5 Ma ice is a challenging task for which this study anticipates and addresses some potential obstacles. The artificial records in Section 5 test a variety of possible scenarios, but an actual Oldest Ice dust log may contain features that cannot be predicted using existing data or thoroughly examined with hypothetical artificial records. In particular, the possibility of folded ice poses a challenge for both site selection and eventual dating and climate interpretation of an Oldest Ice core. Artificial record 4, which simulated folded ice, did have a lower correlation coefficient compared to artificial record 3, but on its own, a single optical log of folded ice would not have a point of reference to make this comparison. One of the advantages of rapid access drilling and optical logging is that it enables multiple boreholes to be excavated and logged across an area of interest. These optical logs could then be compared to each other as a check for folded ice.

Overall, our study comparing marine dust records and peak alignment techniques demonstrates that the very same climate features that motivate interest in the MPT and the 40 kyr world require us to be cautious and critical in applying tools we have used for the 100 kyr world. Until recently, the ODP1090 dust record was the only 1.5 million year old record in this region, and it has been used as a representative Southern Ocean dust signal with implications for biogeochemical cycling, justified by strong correlation with EDC dust over the past 800 kyr (Martínez-Garcia et al., 2011; Chalk et al., 2017). Yet the decreasing

correlation strength with the new U1537 dust record beyond 800 ka suggests that the dust signal may be spatially variable across the mid-to-high latitude southern hemisphere in the 40 kyr world. We note that high amplitude glacial dust peaks in U1537 back to 1.5 Ma complicate the hypothesis that increasing dust flux across the MPT contributed to increasing iron fertilization, $CO_2$ drawdown, and intensification of glacial cycles (Martínez-Garcia et al., 2011; Chalk et al., 2017).

This spatial variability in the marine dust signal may provide insight into early Pleistocene climate dynamics. The Southern Hemisphere westerlies, which transport dust from South American dust sources eastward to site ODP1090 and southeast to site U1537 and the Antarctic plateau, shift equatorward during cold periods and poleward during warm periods (Lee et al., 2011; Vanneste et al., 2015). The less intense glaciations of the 40k world may have resulted in less intensification and equatorward movement of the westerlies during glacial periods, with less dust transport at the lower latitude of ODP1090. This

may contribute to the irregular pre-MPT dust signal in ODP1090 while distinct 40 kyr peaks are still present in U1537 that correspond to cycles in the LR04 benthic $\delta^{18}O$ stack (Fig. 1). Dust transport from South America to the eastern Atlantic Ocean may have also been suppressed under less glacial wind intensification and a warmer, wetter background state due to the 11,000 km distance, while transport to the much more proximal U1537 site was unaffected. Extending the Southern Ocean dust record further back in time at multiple sites would shed light on these spatial patterns, and modelling studies could help address the

underlying dynamical questions.

**6 Conclusions**

We have explored the use of marine records as stratigraphic templates for dating Oldest Ice, including the new U1537 marine dust record that is located closer to a primary dust source than previous records. We have shown that the U1537 dust record is

particularly promising as a reference record resembling the expected pattern of Antarctic dust for the target period 800 ka -1.5 Ma. Using a suite of artificial dust records, we have demonstrated that offset age tie points between an Oldest Ice optical dust log and marine templates may be detectable by comparing the correlation strength of alternate potential alignments with the marine template. These marine dust templates are useful tools for rapid dating of potential Oldest Ice sites as well as eventually dating a recovered Oldest Ice core in conjunction with other methods such as argon isotope dating of deep ice (Bender et al.,

2008; Yan et al., 2019).

The rapid access optical dust logging technologies that are being developed for Oldest Ice site reconnaissance may further be used to evaluate the coherence of the dust signal across the East Antarctic Plateau in the 40 kyr world, independent of Oldest Ice exploration efforts. If the dust signal is consistent, as it is across multiple ice cores the 100 kyr world, it could be used to

identify and reconstruct disturbed deep Oldest Ice cores that have been affected by folding. On the other hand, spatially variable dust signals may reveal differences in atmospheric circulation during the 40 kyr world and the MPT compared to the 100 kyr

world. Disagreement between the U1537 and ODP1090 marine dust records already indicates that we have much to learn about the MPT and the 40 kyr world from paleoclimatic data. To this end, rapid dating of optically logged Antarctic dust records provides a useful tool in the effort to recover an Oldest Ice core.

## Code Availability

Code used in this study is available upon request.

## Data Availability

EDC optical dust log data is available at [uploading to repository in progress]. All other data was previously published.

## Competing Interests

The contact author has declared that none of the authors has any competing interests.

## Author Contribution

Jeffrey Severinghaus and Ryan Bay conceptualized the study, with new direction from Jessica Ng. Jessica Ng developed the dust record comparison methodology and performed the analysis. Ryan Bay and Delia Tosi produced the EDC optical dust log. Jessica Ng prepared the manuscript with contributions from all co-authors.

## Acknowledgments

We are grateful to Sarah Aarons, Michael Weber, and Eric Wolff for helpful conversations that contributed to this study, the EPICA collaboration, the field teams that supported RAID and IceDiver testing, and the many researchers who produced the data sets used in this study. This research was supported by NSF Awards #1419979 and #2019719.

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

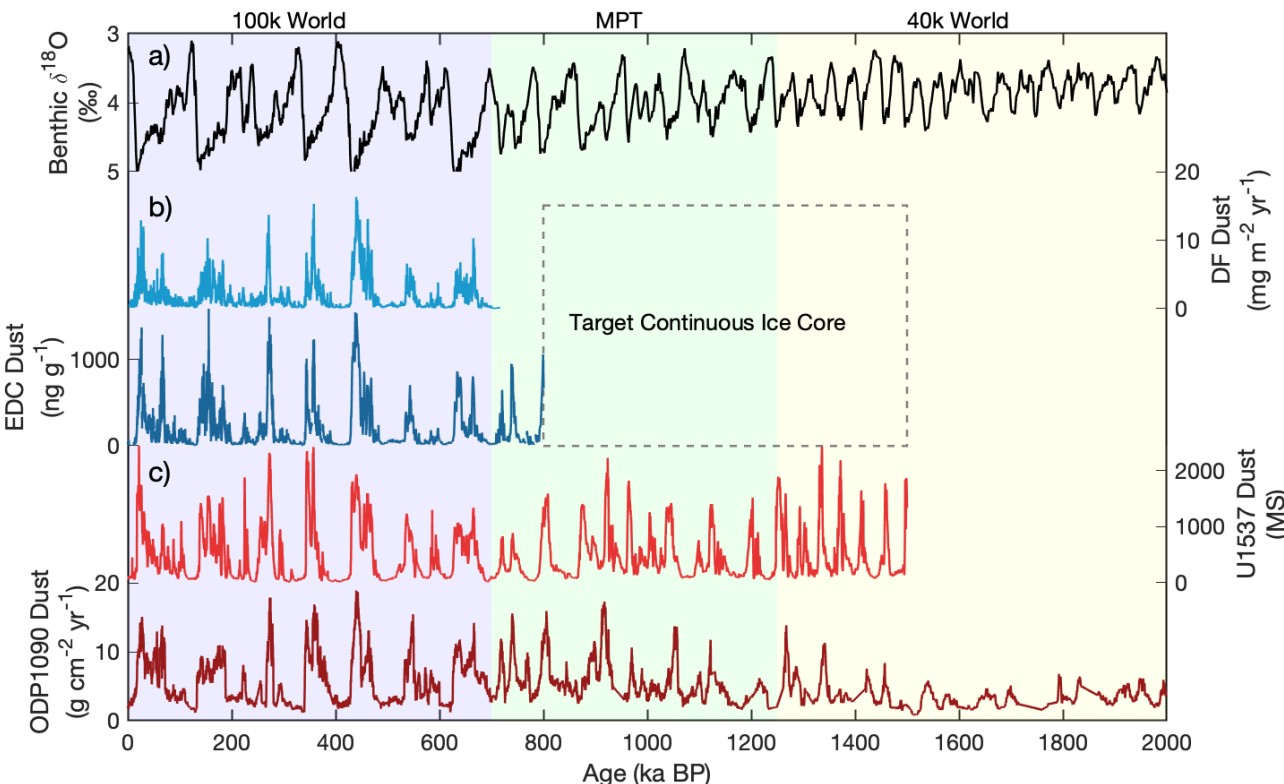

**Figure 1: Time series of LR04 stack of benthic δ¹⁸O, ice dust records, and marine dust records used in this study shown to 1.5 Ma, the target Oldest Ice age. Colored panels indicate the 40k world (yellow), the Mid-Pleistocene Transition (green), and the 100k world (blue).**

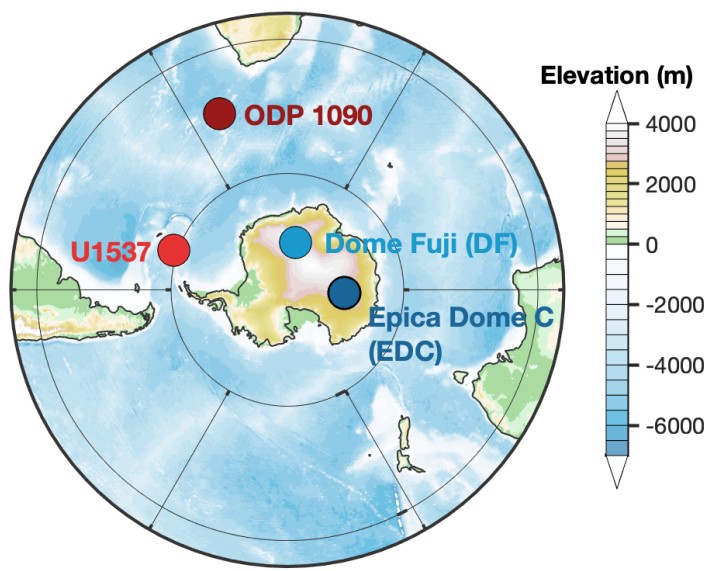

**Figure 2: Location of the four dust records used in this study. EDC (dark blue) was drilled at 75.1 °S, 123.35 °E; DF (light blue) at 77.31 °S, 39.70 °E; U1537 (red) at 59.11 °S, 40.91 °W; and ODP1090 (dark red) at 42.91 °S, 8.90 °E.**

**Table 1: Correlation coefficient R of ice dust, marine dust, and benthic $\delta^{18}O$ records over the common period 3.3ka – 715.9 ka. *Benthic $\delta^{18}O$ was correlated with the common logarithm of the dust records, which is more similar in shape to benthic $\delta^{18}O$ than the original dust records.**

|  | DF | EDC | U1537 | ODP 1090 | LR04* |
|---|---|---|---|---|---|
| DF | x | 0.83 | 0.76 | 0.72 | 0.64 |
| EDC | 0.83 | x | 0.85 | 0.79 | 0.73 |
| U1537 | 0.76 | 0.85 | x | 0.78 | 0.76 |
| ODP1090 | 0.72 | 0.79 | 0.78 | x | 0.76 |
| LR04* | 0.64 | 0.73 | 0.76 | 0.76 | x |

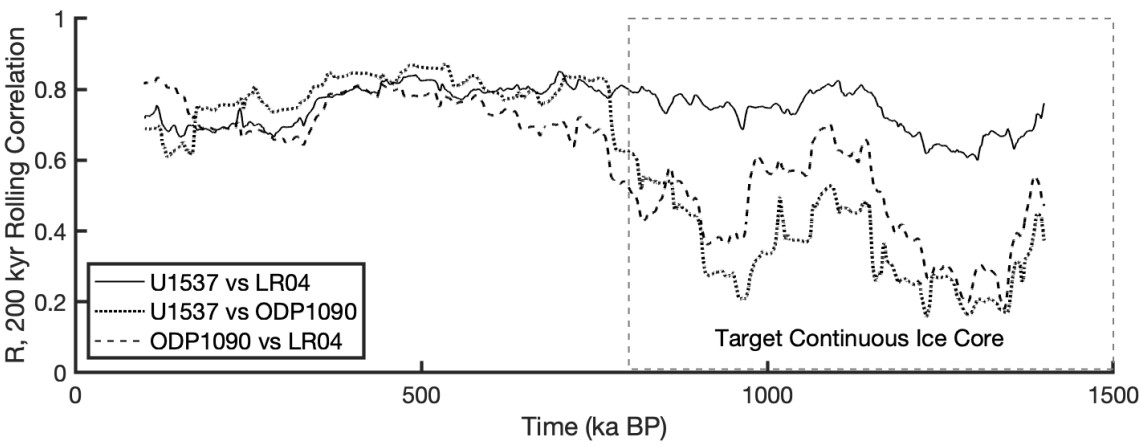

**Figure 3: 200 kyr rolling correlation of marine records with the first and last 100 kyr removed to eliminate edge effects: U1537 vs ODP1090 dust (dotted line), LR04 benthic δ¹⁸O vs logarithm of ODP1090 dust (dashed line), and LR04 benthic δ¹⁸O vs logarithm of U1537 (solid line).**

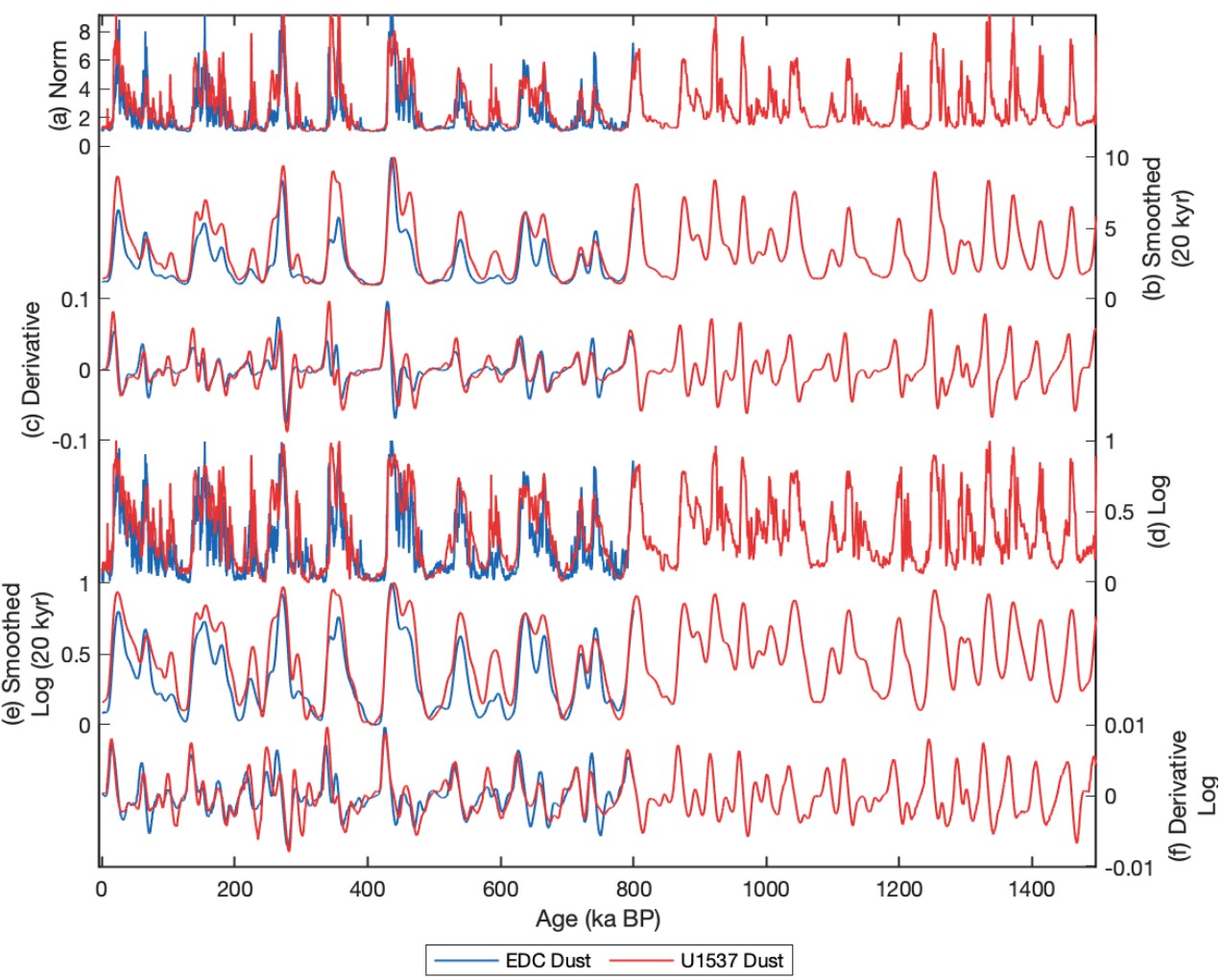

**Figure 4: Time series of EDC ice dust (blue) and U1537 marine dust (red) applying techniques to aid visual pattern matching. a) Normalized EDC and U1537 dust; b) smoothed EDC and U1537 dust; c) derivative of smoothed EDC and U1537 dust; d) logarithm of EDC and U1537 dust; e) smoothed logarithm of EDC and U1537 dust; f) derivative of smoothed logarithm of EDC and U1537 dust.**

**Table 2: Correlation coefficient R of smoothed and otherwise treated EDC and U1537 dust records.**

**EDC vs U1537**

|   | Norm | Smooth | Derivative | Log | Smooth Log | Derivative Log |
|---|------|--------|------------|-----|------------|----------------|
| **R** | 0.74 | 0.91 | 0.86 | 0.87 | 0.93 | 0.85 |

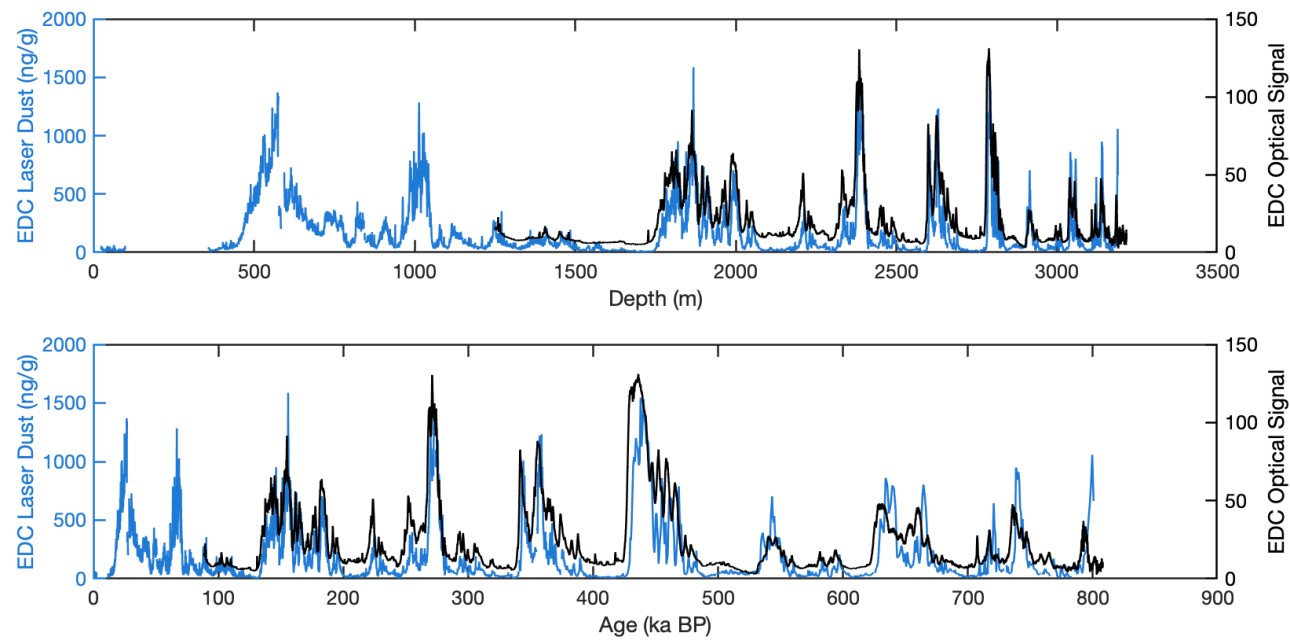

**Figure 5: Comparison between EDC dust measured by laser scattering of the ice core (blue) and optical borehole logging of the borehole (black) over depth (top) and age (bottom).**

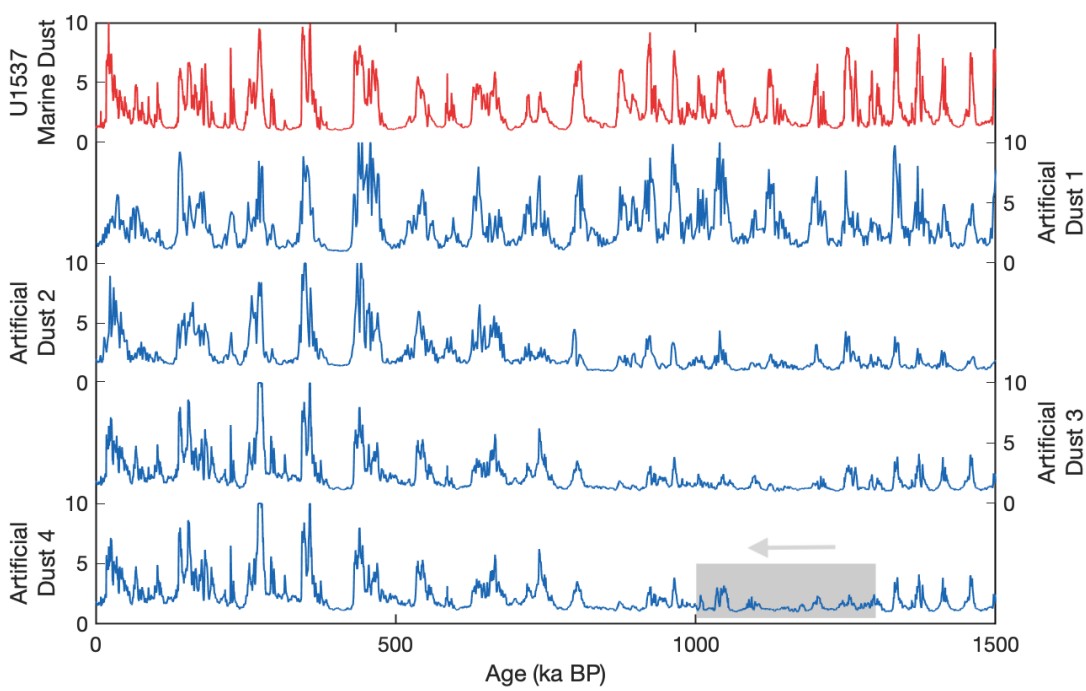

**Figure 6: Artificial ice dust records (blue) created by re-scaling and adding noise to the U1537 marine dust record (red). The amplitude of artificial records 2-4 was reduced over 800 ka – 1.5 Ma, and the gray section from 1 Ma – 1.3 Ma was inverted for artificial record 4 to simulate folded ice, indicated by the arrow.**

**Table 3: Correlation coefficient R of artificial dust records with U1537.**

| ID | Treatment of 800 ka – 1.5 Ma | R | | |
|----|------------------------------|------|-----------|----------------|
|    |                              | All  | 0 – 800 ka | 800 ka – 1.5 Ma |
| 1  | None                         | 0.71 | 0.72      | 0.70           |
| 2  | Decrease amplitude           | 0.62 | 0.82      | 0.75           |
| 3  | Decrease amplitude           | 0.69 | 0.88      | 0.83           |
| 4  | Decrease amplitude, invert   | 0.67 | 0.88      | 0.72           |

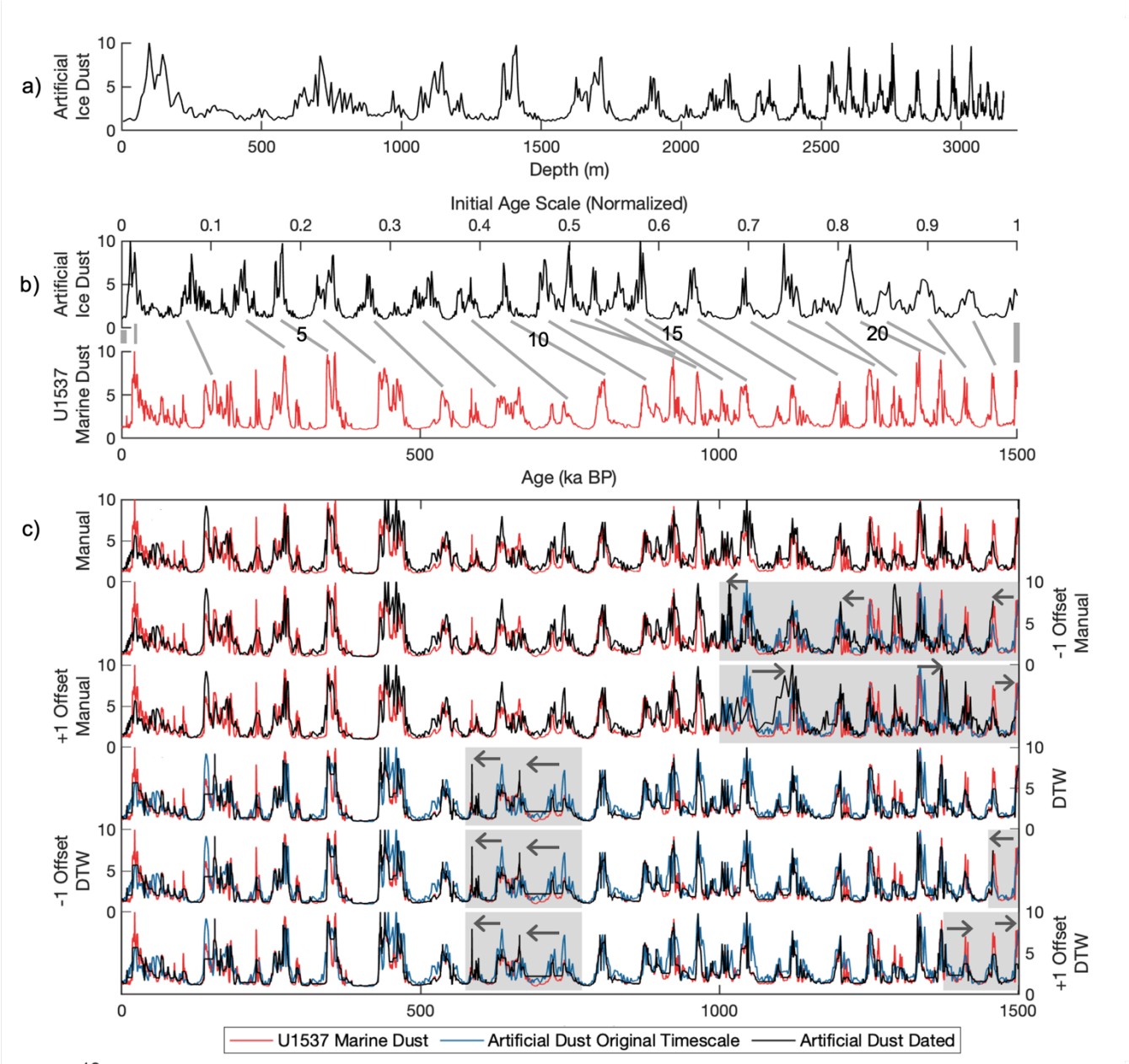

**Figure 7. Dating of artificial Oldest Ice optical dust log. a) Artificial Oldest Ice dust-depth record constructed using artificial record 1 and Nye ice flow model. b) Numbered tie points between artificial record on a preliminary age scale using Nye model (top, black) and U1537 (bottom, red). Start and end tie points are used for dynamic time warping. c) Comparison of manual (upper 3) and DTW (lower 3) dating results (black) to original age scale of the artificial record (blue) and U1537 (red). Shaded panels mark time periods with representative misaligned age tie points indicated by gray arrows.**

**Table 4. Correlation coefficient R of U1537 and artificial Oldest Ice dust record dated by manual peak alignment and linear interpolation (Manual) and dynamic time warping (DTW) with peak alignment errors. +1 and +2 indicate offsets such that the oldest artificial Oldest Ice age appears too old; -1 and -2 indicate offsets such that the artificial record appears too young.**

| Artificial Record | Dating Method | R | | | | |
|---|---|---|---|---|---|---|
| | | Offset -2 | Offset -1 | Offset 0 | Offset +1 | Offset +2 |
| 1, Norm | Manual | 0.61 | 0.64 | 0.72 | 0.65 | 0.67 |
| 2, Norm | Manual | 0.58 | 0.57 | 0.60 | 0.57 | 0.57 |
| 3, Norm | Manual | 0.57 | 0.59 | 0.62 | 0.60 | 0.59 |
| 4, Norm | Manual | 0.57 | 0.58 | 0.61 | 0.59 | 0.59 |
| 1, $Log_{10}$ | Manual | 0.69 | 0.70 | 0.77 | 0.71 | 0.71 |
| 2, $Log_{10}$ | Manual | 0.53 | 0.52 | 0.57 | 0.52 | 0.51 |
| 3, $Log_{10}$ | Manual | 0.55 | 0.57 | 0.60 | 0.56 | 0.55 |
| 4, $Log_{10}$ | Manual | 0.53 | 0.55 | 0.59 | 0.55 | 0.56 |
| 1, Norm | DTW | 0.96 | 0.97 | 0.97 | 0.97 | 0.97 |