# Peer review of "Evaluating marine dust records as templates for optical dating of Oldest Ice"

_EGUsphere, 2023_

## Referee Comment (RC1)

This paper is a rather straightforward extension of previous work that suggested using dust records to date new, old, ice cores. For complete transparency, I was the lead author of that previous paper.

The most important addition of this paper is comparison with a new marine dust record which shows a different pattern in the pre-MPT era and therefore suggests a different template to that proposed so far. This is certainly worth pointing out and the authors make a good, though not cast-iron, case that the new record would be a better template. The second addition is the use of dust data obtained by downhole optical logging, opening up the possibility to date ice rapidly and before a full drilling campaign. This again seems valuable, and is perhaps underplayed in this paper. Finally the authors discuss how the dust templates might be used in practice – this section is less insightful, and quite hard to follow. Taking the paper as a whole, it is worthwhile (and I thank the authors for keeping it short in proportion to its findings), but does need some clarifications and minor additions.

Larger issues:

Line 135-140. Clearly the two marine dust records do diverge considerably before 800 ka. While you present reasons why 1537 should be the better template, I'd like to see a more subtle discussion. I think one could equally make the argument the other way, that 1090 more obviously integrates dust emissions from across South America, while 1537 is likely only to see the southernmost Patagonian emissions.  Dust geochemistry does suggest a preponderance of Patagonian emissions in EDC dust, but it is not really clearcut, and who knows if this is true before 800 ka? I am not arguing for a major change but I think it would be better to leave it slightly open as to which dust record is the better template and let us decide once we have the ice!  I do think you could though make more of the fact that 1537 is much better resolved (I think ~200 m of core for 1.5Ma cf 40 m at 1090).

Line 141-9. I think it is a bit dangerous to imply that you expect LR04 to look like dust. The point about using dust records as a template (and other records we might choose as templates) is that there is a good theoretical reason (same source and transport pathways) to expect them to look the same under most circumstances. This was our justification for proposing extending the match beyond 800 ka (even if 1537 suggests that may also be tricky). There is no similar theoretical reason (other than that all records show glacial cycles) to expect LR04 to look similar, or to propose that the match should extend into the 40 ka world, and using it as a template would be circular reasoning when we want to use the ice record to understand climate. By all means point out the similarities but then I would strongly recommend not using LR04 any further and not recommending that it forms any part of a template. In particular I would not show it in Fig 4.

Optical dust record, section 4.1 and Fig 5. To my knowledge this is the first time the EDC optical dust record has been shown, so you need to do more to show that it matches the laser dust record. Before playing with age scales, you should show both on a depth scale where there is no alignment issue. I suggest adding a figure where you show the whole record and some detailed parts so the reader can judge to what extent the optical record captures both the shape and amplitude of dust peaks. (Minor point: if this is indeed the first outing for these data then there should be an acknowledgment to EPICA!).  In addition to this Fig 5 is confusing: panel a and b say they are the optical log but they have a colour that says (legend) they are laser dust. Please alter this.

Section 4.2 and Fig 6. I have now read this several ties and I'm afraid I can't understand what you have done. "such that the peaks in the artificial record (Fig 6c top) appeared older than they originally were". I just can't see in the figure what it is you claim to have done, or where the supposed mismatches are. You seem to have exactly the same peaks in exactly the same places.  You appear to be suggesting that the peak at 1050 ka is displaced by 200 ka, but I am not seeing it. I

imagine I have misunderstood the figure but I think others will too, so please make a new attempt to explain this perhaps highlighting using the curves in b and c exactly what the mismatches are.

Honestly I don't find section 4.2 very useful or enlightening, but I think this is because I am imagining a situation where we have a core with multiple dating aids (gases with insolation cycles, dust, 10Be etc), while you are considering the case of a raid access hole with only the dust record available. I think the more likely problem for a core is not that peaks in good order are misidentified: the issue is how to know whether there has been folding putting peaks in the ice in the wrong order, and this is not addressed here. Maybe this could be mentioned as a potential hazard!

Detailed comments

Line 12. I am not sure why "surprisingly" is used. At EDC at least there is melting at the bed so it's not a surprise that it's not so old. Perhaps just remove this word unless you had something specific in mind.

Line 10 and 47. Sorry to be picky but IPICS is International Partnerships (plural) in Ice Core Sciences.

Line 90 (also 106). It's a shame you are using the very old EDC3 age model, especially as you use the AICC2012 alignment for Dome Fuji. I appreciate that ODP1090 was compared to an EDC3 age model as that was still the standard in 2012, but it should be quite straightforward to translate both EDC and EDC1090 to AICC2012, thus removing one unintended source of minor mismatch between the records you use. If you do stay with EDC3 perhaps you need to add a line pointing out that there are minor differences between EDC3 and AICC2012, but that they don't affect the pattern of glacial cycles being used in the template.

Line 113: "the aridity (due to temperature) and circulation of the atmosphere, which influences the production and transport of dust". I suggest adding "and atmospheric lifetime". This is probably more important than the transport itself.

Section 2.4, Table 1, etc. You use dust itself in 1090, whereas our previous paper used Fe_MAR. I think your dust record is OK, and it seems to give a good result in the last 800 kyr, and a very similar pattern before that. But it would be good if you just point this difference out so people can understand why the units are a little different. Similarly (and I know you have discussed this) it might need an extra line in 2.3 to explain that MS has been empirically shown to be more like EDC and ODP1090 than what might seem more direct measures of dust.

Line 155. Of course the reason why the log records work better is because the dynamic range of the different records is different. (I think less so for EDC and 1537 but still the principle holds). Add a few words?

Fig. 5 – see comments above about the colours in panels a and b, which are confusing. In addition it should not go beyond 800 ka in panels b and c – this is as far as the EPICA age scale goes and beyond that it is assumed that the ice is disturbed and not necessarily in age order.

Lines 185-6. I found this confusing, and couldn't quite work out what you did (especially what you mean by "scaling the smoothed record by random factors between 0.4 - 0.7 linearly interpolated between 500 kyr intervals"), although I understand the intention. Please spell it out more clearly. I wonder why you smoothed with a 20 kyr running mean – this means that by design you have taken out some of the multimillennial features that might have been used to identify the correct peaks to tie records together.

Discussion: Last para of discussion, you might add some thoughts about dating with multiple datasets in a full core and about the hazards of folding, as per my earlier comment.

Line 265. I'm sorry but I think the optical dust record needs to be made publicly available, not just on request, to meet the journal rules.

---

## Author Comment (AC1)

This paper is a rather straightforward extension of previous work that suggested using dust records to date new, old, ice cores. For complete transparency, I was the lead author of that previous paper.

The most important addition of this paper is comparison with a new marine dust record which shows a different pattern in the pre-MPT era and therefore suggests a different template to that proposed so far. This is certainly worth pointing out and the authors make a good, though not cast-iron, case that the new record would be a better template. The second addition is the use of dust data obtained by downhole optical logging, opening up the possibility to date ice rapidly and before a full drilling campaign. This again seems valuable, and is perhaps underplayed in this paper. Finally the authors discuss how the dust templates might be used in practice – this section is less insightful, and quite hard to follow. Taking the paper as a whole, it is worthwhile (and I thank the authors for keeping it short in proportion to its findings), but does need some clarifications and minor additions.

Thank you for your thoughtful assessment. We will further emphasize the rapid dating application of our approach where appropriate throughout the paper and revise the section on using the dust templates for clarity.

Larger issues:
Line 135-140. Clearly the two marine dust records do diverge considerably before 800 ka. While you present reasons why 1537 should be the better template, I'd like to see a more subtle discussion. I think one could equally make the argument the other way, that 1090 more obviously integrates dust emissions from across South America, while 1537 is likely only to see the southernmost Patagonian emissions. Dust geochemistry does suggest a preponderance of Patagonian emissions in EDC dust, but it is not really clearcut, and who knows if this is true before 800 ka? I am not arguing for a major change but I think it would be better to leave it slightly open as to which dust record is the better template and let us decide once we have the ice! I do think you could though make more of the fact that 1537 is much better resolved (I think ~200 m of core for 1.5Ma cf 40 m at 1090).

We appreciate this nuance and will rework the section accordingly.

Line 141-9. I think it is a bit dangerous to imply that you expect LR04 to look like dust. The point about using dust records as a template (and other records we might choose as templates) is that there is a good theoretical reason (same source and transport pathways) to expect them to look the same under most circumstances. This was our justification for proposing extending the match beyond 800 ka (even if 1537 suggests that may also be tricky). There is no similar theoretical reason (other than that all records show glacial cycles) to expect LR04 to look similar, or to propose that the match should extend into the 40 ka world, and using it as a template would be circular reasoning when we want to use the ice record to understand climate. By all means point out the similarities but then I would strongly recommend not using LR04 any further and not recommending that it forms any part of a template. In particular I would not show it in Fig 4.

This is a good point, although LR04 has already been used to establish the U1537 chronology (along with biostratigraphy and magnetostratigraphy). We will remove LR04 from Fig 4 and most of the analysis.

Optical dust record, section 4.1 and Fig 5. To my knowledge this is the first time the EDC optical dust record has been shown, so you need to do more to show that it matches the laser dust record. Before playing with age scales, you should show both on a depth scale where there is no alignment issue. I suggest adding a figure where you show the whole record and some detailed parts so the reader can judge to what extent the optical record captures both the shape and amplitude of dust peaks. (Minor point: if this is indeed the first outing for these data then there should be an acknowledgment to EPICA!). In addition to this Fig 5 is confusing: panel a and b say they are the optical log but they have a colour that says (legend) they are laser dust. Please alter this.

We will add a figure and more discussion of the EDC optical dust record with appropriate acknowledgement to EPICA. We apologize for the error in the legend and will correct it for the revised manuscript.

Section 4.2 and Fig 6. I have now read this several ties and I'm afraid I can't understand what you have done. "such that the peaks in the artificial record (Fig 6c top) appeared older than they originally were". I just can't see in the figure what it is you claim to have done, or where the supposed mismatches are. You seem to have exactly the same peaks in exactly the same places. You appear to be suggesting that the peak at 1050 ka is displaced by 200 ka, but I am not seeing it. I imagine I have misunderstood the figure but I think others will too, so please make a new attempt to explain this perhaps highlighting using the curves in b and c exactly what the mismatches are.

If DTW were totally accurate, the black and blue lines would lie exactly on top of each other, with the same amplitudes at each peak. However, as an example, the highest amplitude peak in the black line (on the DTW timescale) within the orange shaded area occurs at appx 1200 ka, while the highest amplitude peak in the blue line (on the original time scale) occurs at appx 1050 ka. We will rework this section and figure for clarity with additional suggestions from reviewer Lorraine Lisiecki.

Honestly I don't find section 4.2 very useful or enlightening, but I think this is because I am imagining a situation where we have a core with multiple dating aids (gases with insolation cycles, dust, 10Be etc), while you are considering the case of a raid access hole with only the dust record available. I think the more likely problem for a core is not that peaks in good order are misidentified: the issue is how to know whether there has been folding putting peaks in the ice in the wrong order, and this is not addressed here. Maybe this could be mentioned as a potential hazard!

Our approach is really intended for rapid dating in the field using only the dust record, as you note. We will emphasize this application in the introduction and where appropriate throughout the manuscript. Per your and Lorraine Lisiecki's suggestion, we will explore whether folding could be detected using our method.

Detailed comments

Line 12. I am not sure why "surprisingly" is used. At EDC at least there is melting at the bed so it's not a surprise that it's not so old. Perhaps just remove this word unless you had something specific in mind.

Removed "surprisingly."

Line 10 and 47. Sorry to be picky but IPICS is International Partnerships (plural) in Ice Core Sciences.

Corrected.

Line 90 (also 106). It's a shame you are using the very old EDC3 age model, especially as you use the AICC2012 alignment for Dome Fuji. I appreciate that ODP1090 was compared to an EDC3 age model as that was still the standard in 2012, but it should be quite straightforward to translate both EDC and EDC1090 to AICC2012, thus removing one unintended source of minor mismatch between the records you use. If you do stay with EDC3 perhaps you need to add a line pointing out that there are minor differences between EDC3 and AICC2012, but that they don't affect the pattern of glacial cycles being used in the template.

We will put EDC and ODP1090 on the AICC2012 chronology.

Line 113: "the aridity (due to temperature) and circulation of the atmosphere, which influences the production and transport of dust". I suggest adding "and atmospheric lifetime". This is probably more important than the transport itself.

Added "and atmospheric lifetime."

Section 2.4, Table 1, etc. You use dust itself in 1090, whereas our previous paper used Fe_MAR. I think your dust record is OK, and it seems to give a good result in the last 800 kyr, and a very similar pattern before that. But it would be good if you just point this difference out so people can understand why the units are a little different. Similarly (and I know you have discussed this) it might need an extra line in 2.3 to explain that MS has been empirically shown to be more like EDC and ODP1090 than what might seem more direct measures of dust.

We will clarify these details, thank you for pointing them out.

Line 155. Of course the reason why the log records work better is because the dynamic range of the different records is different. (I think less so for EDC and 1537 but still the principle holds). Add a few words?

We will add a brief comment on the log records.

Fig. 5 – see comments above about the colours in panels a and b, which are confusing. In addition it should not go beyond 800 ka in panels b and c – this is as far as the EPICA age scale goes and beyond that it is assumed that the ice is disturbed and not necessarily in age order.

We apologize for the error in the legend colors that caused confusion. The exercise shown in Fig 5 pretends that we only have the dust-depth record, not an established age scale, as would be the case for an Oldest Ice site; therefore we do not limit the age of U1537 marine dust shown in panel C to only 800 ka.

Lines 185-6. I found this confusing, and couldn't quite work out what you did (especially what you mean by "scaling the smoothed record by random factors between 0.4 - 0.7 linearly interpolated between 500 kyr intervals"), although I understand the intention. Please spell it out more clearly. I wonder why you smoothed with a 20 kyr running mean – this means that by design you have taken out some of the multimillennial features that might have been used to identify the correct peaks to tie records together.

We will revise this section for clarity, as well as implement suggestions from Lorraine Lisiecki to reduce the amplitude of the artificial record farther back in time and compare multiple versions of an artificial record. The thought behind smoothing with a 20 kyr running mean was that some lower amplitude millennial scale features that are recorded in marine sediments may not make it to the ice sheet, but we will try it both with and without this smoothing.

Discussion: Last para of discussion, you might add some thoughts about dating with multiple datasets in a full core and about the hazards of folding, as per my earlier comment.

Per our response to the earlier comment, we will assess whether our approach can detect folding and add to the discussion here.

Line 265. I'm sorry but I think the optical dust record needs to be made publicly available, not just on request, to meet the journal rules.

We will make the optical dust record available in a data repository.

---

## Author Comment (AC2)

This manuscript is quite practically focused on initial assessment of drilling locations for oldest ice, proposing and demonstrating that borehole optical logs of dust would be an effective method to estimate the age of ice back to 1.5 Ma. Overall, I find this to be a valuable scientific contribution worthy of publication. However, a few portions of the manuscript need to be clarified and a few more calculations/experiments would also be very useful.

We appreciate this evaluation and the very useful suggestions offered.

Needed clarifications:
1. Line 117-118: The meaning here is ambiguous about what the authors did to "correlate" the records and put them "on a common timescale." I think the intended meaning is that the records were sampled at common time steps in order to calculate the correlation coefficient between the signals. However, the phrasing could alternatively be interpreted as *stratigraphic* correlation that aligned the records to one common target timescale. Please clarify. If the records are all staying on their respective published timescales described in section 2, then a brief discussion of the extent to which these timescales are expected to agree or differ is warranted.
    A. Your reading is correct: the records were sampled at common time steps on their respective timescales. The sentence in lines 117-118 was meant to introduce the paragraph, not imply that the records were aligned to one common target time scale. We will reword this section to avoid confusion.
2. Like Eric Wolff, I found Figure 5 to be extremely confusing. The caption, axes labels, and legend don't seem to agree with one another. I think the blue line color was used for two different types of data. If so, the problem could be fixed by changing the line color for laser dust in panel c to another color that isn't already used for something else.
    A. We apologize for the error in the legend color that caused this confusion and will correct it in the revised manuscript. Per
3. The description of how the artificial dust record was created is not clear, particularly "scaling the smoothed record by random factors between 0.4 - 0.7 linearly interpolated between 500 kyr intervals" (line 186). An additional concern I have about this methodology is that the laser dust record for EDC shows a much weaker dust amplitude from 500-800 ka than 0-500 ka. Are depth-dependent processes contributing to the weakening of the dust signal in the ice? If so, it would be more realistic if your artificial borehole dust record progressively decreased in amplitude farther back in time rather than scaling randomly through time.
    A. We appreciate this suggestion. We will reduce the dust amplitude of the artificial record farther back in time and clarify the description.
4. It is very difficult to visually discern misalignment in the DTW results, which strengthens the manuscript's conclusion that DTW is not a suitable method for initial comparison between the borehole optical dust logs and the marine record. This point would be strengthened if the authors could find an additional way to illustrate the misalignment. For example, at the top of panel c, the artificial dust record could be plotted on its original, correct age model and arrows could be drawn to DTW results to show how dust peaks were misaligned.

      A. The misalignment has been difficult to visualize so we appreciate this suggestion and will try it out.

Additional calculations/experiments:
1. The weaker correlation for derivatives of the log of the data than the log itself is worthy of a bit more discussion/exploration. I wonder whether this might be an artifact of small differences in the age models for each record. The derivative has higher frequency variability and is therefore more likely to be affected by small age model differences/misalignments. This might actually make the derivative more useful for alignment than the log itself. The correlation of the derivative may also be advantageous if it is less sensitive to long-term trends in the mean or amplitude of the signal through time. This could potentially be evaluated by improving the alignment between the records or by analyzing synthetic records.
    A. The sensitivity of the derivative to higher frequency variability might make it more useful for fine-scale alignment, but perhaps not for alignment on the scale of glacial cycles, which is our focus for this rapid dating method intended for use in the field. We will mention this in the discussion.
2. Was the alignment experiment for an artificial dust signal repeated multiple times with different random noise or only once? Multiple iterations would be useful for characterizing the probability of incorrect conclusions. Additionally, what if the correlation between UF1537 and dust ice were slightly lower before 800 ka (e.g., R=0.7 instead of 0.76)?
    A. Great suggestions. We will implement sensitivity tests for the artificial dust signal with slightly weaker correlation in the older part of the record.
3. Eric Wolff makes a good point about the need to detect ice disturbances that could produce repeated or jumbled sections of the dust time series. I recommend creating an artificial signal that contains jumbled or duplicated sections (from the modified marine dust record) to determine whether your proposed method would detect a misfit between the simulated disturbed ice and the original marine dust record.
    A. This is also a good suggestion and we will try it.

---

## Author Comment (AC3)

In this manuscript, Jessica Ng et al. present a new marine dust record near South America which could be used as a target to date the forthcoming Oldest Ice ice core dust records from Antarctica. They show that their new U1537 marine dust record differs significantly from the published ODP 1090 dust record for the MPT and pre-MPT periods, 1.5-0.8 Myr ago. They argue their new site is better since the correlation with the LR04 stack stays more or less constant in the pre-MPT period with respect to the post-MPT period, contrary to the ODP 1090 record. They argue that it is possible to measure the dust record in Antarctica by logging the ice borehole, and they present such a new borehole dust logging from EPICA Dome C. They then show simple tuning strategies to quickly come-up with a time scale once the new ice dust record is available.

The paper is generally focused and straightforward, yet important, so it was a pleasure for me to review it.

Thank you for this assessment of the paper's readability and significance and for your helpful suggestions.

I find the new dust record from U1537 very interesting. We can discuss if it is really better than ODP1090 (personnally I am quite convinced), but at the very least it is an alternative record which shows that ODP1090 should be taken with caution.

I do agree with Eric Wolff that the new borehole dust record from EPICA Dome C could get a bit more attention if it is really the first time such record is published. A quick comparison with the Coulter and lazer records would be interesting.

We agree with this emphasis on the new EPICA Dome C record and will add a comparison with the Coulter and laser records.

Regarding the end of the manuscript with the tuning strategy, I think there should be more powerful strategies, this is only a first-step strategy (but I think the authors are honnests in presenting it this way). And I do agree with Eric Wolff that this part is not as well presented and straightforward as the other parts. For example, the authors use a Nye ice flow model, but never explicitly describe the parameters they used, while for example the melting is a primary parameter for dating the old section of an ice core.

I would suggest to use the 1D model used in, e.g., Parrenin et al. (TC, 2017), Lilien et al. (TC, 2021) and Chung et al. (TC, in press), which uses a Lliboutry velocity profile. This model has an analytical thinning function and accounts for temporal variations of accumulation through a simple change of time variable. It should give a far better accuracy, while still being very easy to implement. I do not make a strong requirement to use this model, but I think it would be an improvement and I can provide guidance on request.

We appreciate the suggestion to use a more complex model; however, our approach is to keep things as simple and transparent as possible so that an estimate of the basal age of the ice can be made quickly in the field. We will describe the parameters used in the Nye model and revise the

section for clarity. We will also emphasize that we are presenting a first step strategy and that other more complex methods will need to be used to establish a full chronology.

Moreover, the strategy is presented as decoupled between the modelling and the tuning, while I think both should be coupled: one first tune the top part, then apply the model with these dating constrains to extrapolate, then one tune the following part, etc. I personnally think the best approach would be to adjust the glaciological parameters of an ice core dating model in a Bayesian code like IceChrono/Paleochrono so as to optimize the fit with several targets, using a powerful MonteCarlo approach. (Such a method has been presented in the PhD of Jai Beeman in 2019, but unfortunately it has never been published elsewhere).

Our strategy is meant for rapid dating applications that could be used in the field to estimate the basal age of the ice, so we have kept it as simple as possible. Of course other more complex approaches such as described here should be used to establish a more accurate chronology later.

Minor comments:

- L. 40-45: The discussion of gradual vs abrupt MPT, as presented in Legrain et al. (2023) would fit nicely in this introduction, but I let you decide.

- We will add this reference to the introduction.

- L. ~50: In my opinion, the best evaluation of the age and state of basal ice in the Dome C area is from Chung et al. (TC, in press), but I let you decide if you want to cite it.

- We will add this reference.

- L. 96: If I am correct, DFO2006 is the O2/N2 age scale of the first Dome Fuji ice core, DF1. The age scale from Dome Fuji members (2017) is DFO2006, then extended using AICC2012 for DF2. Please check, I don't think this age scale has a proper name, but I would call it DFO2006+AICC2012.

- We have changed the age scale name to DFO2006+AICC2012.

- L. ~120: In Table 1, the correlation coefficients are given for the ice core dust records, not for their log. The coeffs for the log-records are given a bit later in the manuscript, but not in Table 1. I personnally think the coeffs for the log-records are more relevant than for the raw records and I would put them in Table 1.

- We find the comparison of the original unaltered records to be useful and will consider showing correlation coefficients for both the records and the log-records in Table 1.

---

## Author Response (AR2)

Many thanks for sending me the final answers to reviewers.
The comments of reviewers were properly addressed in the new version of the manuscript.
I am thus pleased to tell you that the manuscript is accepted but I would like you to take into account the following technical corrections:
Thank you very much.

1- line 213 – « the amplitude » of what ?
Changed to "the amplitude of the signal"

2- The use of kyr or ka should be checked – as an example, on line 218, you should use « ka » and not « kyr »
Changed "ka" to "kyr" on line 218 and added a brief definition of "ka" in line 44 at the first usage: "800,000 years ago (800 ka)"

3- Fig 6 – explain in caption the meaning of the grey shaded area and corresponding arrow
Changed to "the gray section from 1 Ma – 1.3 Ma was inverted for artificial record 4 to simulate folded ice, indicated by the arrow."